# Using Language Model to Bootstrap Human Activity Recognition Ambient Sensors Based in Smart Homes

**Damien Bouchabou** [1,2,*], **Sao Mai Nguyen** [1,*], **Christophe Lohr** [1], **Benoit LeDuc** [2] **and Ioannis Kanellos** [1]

1   IMT Atlantique Engineer School, 29238 Brest, France; christophe.lohr@imt-atlantique.fr (C.L.);
    ioannis.kanellos@imt-atlantique.fr (I.K.)
2   Delta Dore Company, 35270 Bonnemain, France; bleduc@deltadore.com
*   Correspondence: damien.bouchabou@imt-atlantique.fr or dbouchabou@deltadore.com (D.B.);
    nguyensmai@gmail.com (S.M.N.)

**Abstract:** Long Short Term Memory (LSTM)-based structures have demonstrated their efficiency for daily living recognition activities in smart homes by capturing the order of sensor activations and their temporal dependencies. Nevertheless, they still fail in dealing with the semantics and the context of the sensors. More than isolated id and their ordered activation values, sensors also carry meaning. Indeed, their nature and type of activation can translate various activities. Their logs are correlated with each other, creating a global context. We propose to use and compare two Natural Language Processing embedding methods to enhance LSTM-based structures in activity-sequences classification tasks: Word2Vec, a static semantic embedding, and ELMo, a contextualized embedding. Results, on real smart homes datasets, indicate that this approach provides useful information, such as a sensor organization map, and makes less confusion between daily activity classes. It helps to better perform on datasets with competing activities of other residents or pets. Our tests show also that the embeddings can be pretrained on different datasets than the target one, enabling transfer learning. We thus demonstrate that taking into account the context of the sensors and their semantics increases the classification performances and enables transfer learning.

**Keywords:** sensors embedding; human activity recognition; deep learning; smart home; ambient assisting living; language model; contextualized model; long short-term memory; LSTM; transfer learning; ambient sensors; Word2Vec; ELMo; semantic model

## 1. Introduction

Recent advances in the Internet of Things (IoT) hardwares, particularly in terms of energy consumption, cost or inter-operationality [1], have boosted the development of smart environments, such as smart homes. With more and more new constructions incorporating smart sensors and actuators, a new field of automated home assistance services is opening up, focusing on improving the life quality, autonomy, health, and well-being in smart homes for the elderly or disabled [2]. Moreover, smart homes can provide many other useful services, such as energy management or security systems. However, in order to offer both automated and customized services, a smart home must be able to understand the daily activities of the residents.

### 1.1. Recognition of Daily Activities in Smart Homes

In general, Human Activity Recognition (HAR) in smart homes consists of classifying streams of sensor traces into Activities of Daily Living (ADLs). These traces are captured by a variety of sensors (motion, open/close door, temperature, etc.) integrated in the environment or in objects of the house [3]. Through the sensor logs, algorithms identify the activities of residents using appropriate HAR techniques. Nevertheless, recognizing ADLs in a smart home is a challenging task. One has to take into account: (1) the structure and the topology of the houses, that are generally different; (2) the diverging equipment

and living habits of the residents; (3) the number of residents living at the same time (the more residents there are, the more sensors activation are mixed into sensors' activations sequences); and (4) the dependence of ADLs on specific contexts related to the room where the activity is performed, to the objects, to the devices of the house used, or even to the type of interactions, during the activity. Moreover, sensors logging human activities in smart homes are event-triggered. This last generates non-regularly sampled and sparse data, comparatively to activity recognition using videos (see Reference [4] for a comparative survey). Therefore, we have to continuously deal with challenges in terms of pattern recognition and temporal sequence analyses [5]. Clearly, distinct recordings of the same activity would show little similarity.

Another issue is related to the dataset itself; indeed, most of the sensor activation data are not annotated. One of the reasons is that the dataset has been built with predefined set of activities/classes to be labeled. Thus, the sensor activations, linked to other activities, are annotated under the label "Other". Event sequences in this by-default class are quite different from each other and may be very similar to event sequences from regular classes. Classification difficulty increases proportionally to this "Other" class growth. For instance, this special class represents more than 50% of studied datasets, provided by the Center of Advanced Studies In Adaptive System (CASAS) [6]. This induces class confusion.

Finally, each sensor activation gives little information about the current activity by itself: for instance, the activation of the motion sensor in the kitchen could indicate activities, such as "cooking", "washing dishes", or "housekeeping". Thus, the information this sensor offers us is exploitable only in conjunction with neighboring sensor activations. Thus, our non-Markovian time series is sparse and irregular, and our dataset contains unbalanced classes and highly variable data.

We propose two methods to automatically encode sensor activation which incorporates information addressing neighboring activations. We explored different types of encoding that take advantage of the co-occurrence of sensor but also their sequential relationship. Using successful embeddings for Natural Language Processing (NLP), we tested and reported results from an encoding based on static word embedding, Word2Vec [7], and an encoding based on contextualized word embedding, Embeddings from Language Model (ELMo) [8].

*1.2. Smart Home Data Embedding*

The data representation needs to find both the "meaning" of individual sensor activation and the "meaning" of the combination of sensors' activations. Such a need splits up into two embeddings: lexical and contextual.

1.2.1. Lexical Level Embedding

Instead of mapping sensor activities into arbitrary and uncorrelated vectors, shrewd embeddings can improve the recognition performance [9–11]. Independently of the id of the sensor and the value of its activation, each sensor type also carries a meaning. The nature of the activated sensor or the type of interaction can, thus, translate different activities. Clearly, a movement does not have the same meaning as the opening of a door. The same applies for sensors located nearby or in the same room. Thus, the activations of sensors of the same or nearby type should be correlated to produce an embedding which takes into account the lexicon and semantics of the sensors types, localization, and activation values.

1.2.2. Context Level Embedding

More than just isolated sensor's activations, the sensors' logs constitute a stream of traces which are correlated with each other forming syntactical patterns. Indeed, a same sensor activation or a same interaction may reflect different activities. For example, a motion caught in the kitchen can be attached to the activity of "cooking" or "washing dishes", etc. If the motion sensor is activated, along with an oven switched on or with

the tap turned on, the meaning is not the same. This illustrates that the meaning of a particular sensor activation depends not only on the identity of the sensor and its value but also on the activations of sensors that surround it at that moment. Beyond a single sensor activity, the recognition of ADL algorithm needs to relate it to temporally close sensor activities and incorporate information of the temporal context. This proximity has to be understood in terms of timestep in the time series. A sensor activation can be linked to another activation $n$ timesteps before or after. Thus, the ADL recognition algorithm should integrate contextual representations.

*1.3. Contributions*

This article aims to address the encoding of a sensor's activation according to its context, in order to improve the classification performance for sparse and unevenly spaced time series. It is somehow close to activity recognition studies based on video, smartphone, or wearable sensors. However, its particularity is that it uses data from home automation sensors, under an NLP paradigm.

In this paper, we propose to examine how automatic embeddings of sensor activations can incorporate their lexical and contextual semantics, and how they can improve the performance of human activity recognition algorithms. In particular, we propose to apply two methods coming, precisely, from the field of NLP: a method for static word embeddings, Word2Vec, and a method for contextualized word embedding, ELMo. We combined Word2Vec and ELMo embeddings with a state-of-the-art HAR algorithms using Long Short Term Memory (LSTM). To our knowledge, it is the first time a semantic analysis is incorporated for activity recognition in smart homes. We show that this combination leads to better recognition performances by reporting the performance of four methods:

1. Simple LSTM or a bidirectional LSTM with a softmax layer.
2. One embedding layer, followed by a simple LSTM or a bidirectional LSTM with a softmax layer (Liciotti et al. [12]).
3. Word2Vec embedding layer, followed by a simple LSTM or a bidirectional LSTM with a softmax layer.
4. ELMo embedding layer, followed by a simple LSTM or a bidirectional LSTM with a softmax layer.

We compare the embeddings obtained by Word2Vec and ELMo, and we show that transfer learning is possible with this approach.

The goal of this study is to classify pre-segmented sequences of smart home sensor data into activities of daily living. Our results show:

- the importance of contextualized semantic representations for activity recognition in smart homes;
- that Word2Vec and ELMo have the ability to extract information and feature from real smart home data, leading to better classification performance; and
- that contextualized embedding can bootstrap transfer learning across smart home datasets and demonstrate robustness to noise in sequences.

To sum up: our contribution demonstrates that, by encoding a sensor activation depending on its context, we can improve the classification performance for sparse unevenly spaced time series.

Our code is open and available at https://github.com/dbouchabou/Using-Language-Model-To-Bootstrap-Human-Activity-Recognition-In-Smart-Homes, accessed on 12 August 2021.

**2. Related Work**

To find a suitable embedding for smart home data that can tackle the challenges described above, we review the methods deployed in ADLs recognition based on pattern recognition and spatio-temporal sequence analysis, as well as the methods used for extracting semantic features coming from the NLP domain.

### 2.1. Pattern Recognition Approaches

To recognize ADLs based on sensor traces, researchers used various machine learning algorithms, as reviewed in Reference [13]. These can be divided into two streams: the algorithms exploiting a spatio-temporal representation, with Naive Bayes, Dynamic Bayesian Networks, and Hidden Markov Models; and the algorithms based on features classification, with Decision Tree, Support Vector Machines, or Conditional Random Fields. These approaches are robust and easy to implement and require little computing power. However, they commonly use handcrafted feature extraction methods. This does not allow these methods to adapt when the topology becomes different or the residents' habits change. They are limited to working only in the environment for which they were designed and do not scale up due to lack of generality.

Automatic feature extraction is one of the challenges addressed by Deep Learning (DL) approaches. Recently, a variety of DL algorithms have been applied for HAR focusing on pattern recognition used Convolutional Neural Networks (CNN). They have three advantages for HAR: (1) they can capture local dependencies, i.e., the importance of neighboring observations correlated with the current event; (2) they are scale invariant in terms of step difference or event frequencies; and (3) they can learn a hierarchical representation of data. Researchers used 2D [14,15] and 1D [16] CNN on HAR in smart homes.

The 2D CNN are used on sensors activity sequences transformed into images [14,15]. This approach obtained good classification results on pre-segmented activity sequences but is not suitable for real-time recognition. Indeed, real-time activity recognition consist of associate an activity label for a current time window. Additional work [17] has been conducted in this direction to tackle this problem but is not robust enough to deal with unbalanced datasets and unlabeled events.

The 1D CNN appears to be a competitive solution on spatio-temporal sequence problems [16]. Most recent work [18] with a more complex 1D CNN structure, a Fully Convolutional Network (FCN), achieved good results. Nevertheless, the activity sequences are of variable length, and it is necessary to complete the sequences with a zero fill to train the algorithms. However, this approach does not manage padding correctly in activity sequences, which reduces classification performance.

CNN models have the advantage to be fast to train and achieve high accuracy, but they cannot use long-term dependencies.

### 2.2. Time Series Approaches

Another stream of DL approaches focused more on the temporal aspect of the data stream LSTM have also led to good performance in HAR in smart homes, as reported in References [12,19].

The work of Singh et al. [19] compares a CNN and a LSTM approaches and demonstrates that LSTM perform better on the classification task because it allows learning of temporal information from the sensor data.

Liciotti et al. [12] compare different LSTM structures to different deep learning and traditional machine learning models on pre-segmented activity sequences. They show that LSTM-based approaches, and particularly bidirectional LSTM, obtain better results, owing to their ability to exploit their internal memories to capture long-term dependencies in variable-length input sequences.

LSTM-based approaches better model order and density of events; thus, they better represent the macro structure of an activity. LSTM seem a viable solution to significantly improve the HAR task in the smart home, despite a longer training time than CNN-based approaches.

### 2.3. Semantic Approaches

The common point of both approaches using DL algorithms is that they extract automatically features from raw data and capture long-term dependencies. However, they ignore the semantics, the sensor type, and the context in which the sensor is activated.

Semantic analysis and context modeling has been the focus of NLP research, where the last breakthroughs proposed different unsupervised pre-training methods for an embedding encapsulating some language model. Word embeddings and language models capture information concerning the construction of words, sentences, and texts. They are able to capture the context of a word in a document, semantic and syntactic similarities, relations with other words, etc.

Several training structures and methods have enabled advances in NLP. They can be classified into static word embedding approaches, such as Word2Vec [7], GloVe [20], FastText [21], etc.; and contextualized word embedding approaches, such as ELMo [8], BERT [22], GPT [23], etc. Among these methods, Word2Vec seems the most renowned static word embedding technique. On the other hand, ELMo has made great progress with the contextual representation of words it proposes. The unsupervised training methods have been shown to be effective in translation, text generation, and classification tasks.

Many works have already investigated the use of word embedding for HAR. For instance, Cao et al. [24] applied the Word2Vec model to cluster and create a semantic relationship between population habits, whereas Matsuki et al. [25], Shimoda et al. [26] used pretrained public word embeddings to associate a label with unknown activities for wearable sensors. Abramova et al. [27] exploited a similar approach to annotate unknown activities and studied the zero-shot learning in a smart home. Nevertheless, to our knowledge, the training of unsupervised learning methods has not been exploited for the problem of classifying ADL sequences in smart homes. Furthermore, capturing the context and semantics of sensor activation is still not exploited today and could improve the performance of ADLs classification algorithms, such as LSTM.

## 3. Background

We describe in the sequel the background that leads to ideas defended in this paper. We detail, in particular, the two embeddings we adapted from the NLP domain: Word2Vec and ELMo. Our work is built up on these proven methods. Our goal is to deviate their use in order to extract contextual and semantic information from smart home sensors.

### 3.1. Word2Vec: Context-Free Embedding

Word2Vec is one of the most popular techniques to learn word embeddings using shallow neural networks. It was developed by Mikolov et al. [7]. Word2Vec is an unsupervised learning technique to learn continuous representations of words. In many ways, Word2Vec builds on a bag of words, but, instead of assigning discrete tokens to words, it learns continuous multi-dimensional vector representation for each word in the training corpus. There exist two main methods for learning representations of words in the Word2Vec algorithm (Figure 1): (1) the Continuous Bag Of Words (CBOW), trained by learning to predict the center word based on the context words, as in Figure 1a; (2) the Skip-Gram method, which is trained to predict, given a target word, the most probable words in a fixed sized window around it, as in Figure 1b.

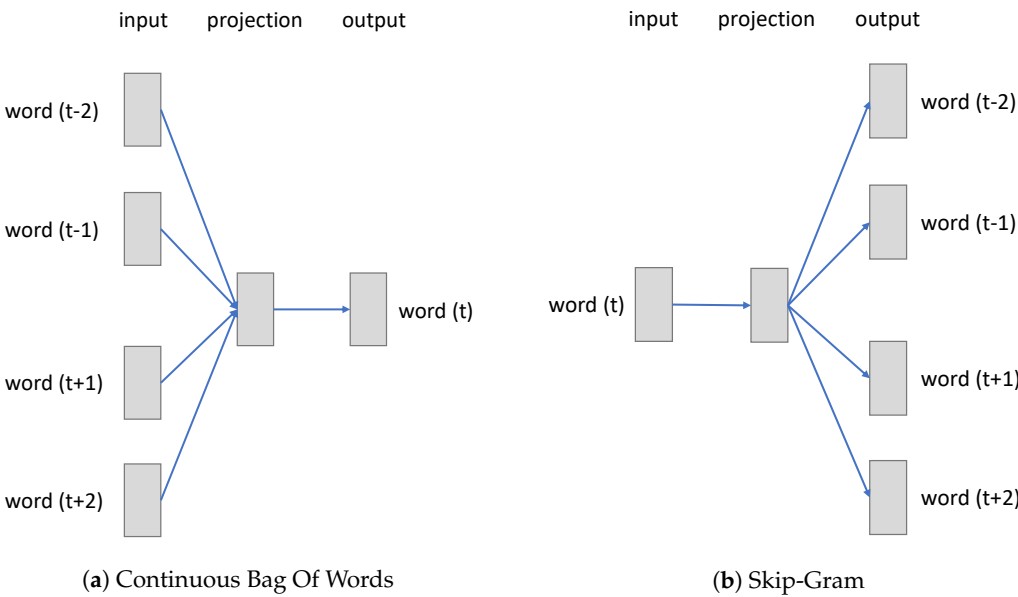

(**a**) Continuous Bag Of Words　　　　　　(**b**) Skip-Gram

**Figure 1.** Word2Vec methods.

By the way it is designed, Word2Vec captures the similarity of words in a corpus. Moreover, thanks to the distance of a word from other words it calculates, Word2Vec also captures some sense of the word.

Word2Vec is a powerful technique. Nevertheless, the main problem of this word embedding is that it provides a single representation for each word regardless of the context. Put differently, the word "orange" in syntagms, such as "the orange juice" or "the orange car", garners the same vector representation, although "orange" does not have the same meaning in the two sequences of words. This lack of context understanding is an important issue to capture the sentence's meaning and introduce the well-known polysemy problem.

### 3.2. ELMo: Contextualized Embedding

Generally, word embeddings, such as Word2Vec, fail to deal efficiently with lexical polysemy. These word embeddings cannot take advantage of information of the context in which the word was used. Instead of using a fixed embedding for each word, ELMo [8] looks at the entire sentence before assigning each word in its embedding. The core idea behind contextual word embeddings is to provide more than one representations for a word, based on the context in which this word appears. ELMo uses a bidirectional LSTM trained on a specific task to be able to create such embeddings. More exactly, it uses two parallel stacked LSTM to capture information from past and future context to encode the current token. In addition, a residual connection is created between the two staked LSTM. ELMo acquires its understanding of language by being trained to: (1) predict the next word in a sequence of words and (2) also predict the previous word. The former task is called "language modeling", and the latter "reverse language modeling". This method of learning is convenient because such a model can learn without the need for labels.

Usually, the ELMo representation is the weighted sum of the outputs of the different layers of the model, where the weights are trained according to the final task (see Figure 2a). However, we also evaluate three other output forms as depicted in Figure 2. First, we use the simple sum of outputs (Figure 2b). Secondly, we evaluated with only the output of the last layer of the model (Figure 2c). Finally, we have found in our experiments that concatenating the output of the different layers of the model provides the best results (Figure 2d). We selected the concatenation output to train the classifier. This output allows the classifier to use different levels of representation of the sequence and select on its own which information in each representation is important to accomplish the final task.

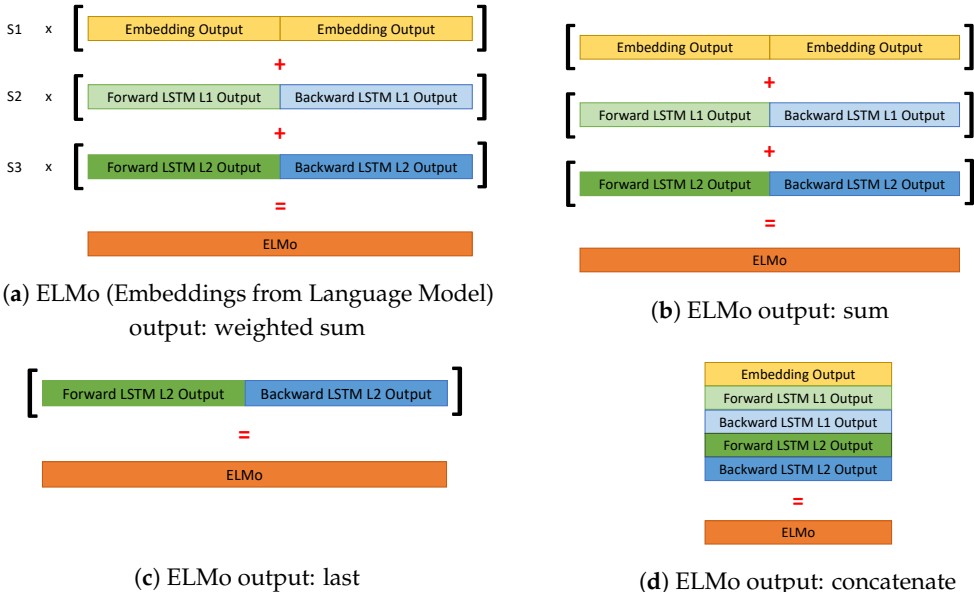

(**a**) ELMo (Embeddings from Language Model) output: weighted sum

(**b**) ELMo output: sum

(**c**) ELMo output: last

(**d**) ELMo output: concatenate

**Figure 2.** ELMo outputs.

## 4. Proposed Approach

### 4.1. Key Ideas

Our approach tackles the contextual representation of each IoT activation by proposing language models of lexical and contextual semantics able to encode the context of sparse and unevenly spaced time series. We argue that HAR algorithms need to use models that can extract and use semantic and contextual information. Indeed, the encoding of the $i$th activation data should not depend only on its value and be expressed as a function $f(activation_i)$. Instead, it should capture more information on sensor behavior, incorporating relevant neighboring activations while smoothing out variability and noise. In other words, the sensor encoding at $i$th activation should also depend on the other activations in the sequence, and be expressed as a function $f(activation_i, \{activation_1, activation_{i-1}\}, \{activation_{i+1}, activation_n\})$, where $n$ is the number of activations in the sequence.

Below, we describe the proposed system for human activity recognition in smart homes, designed as a classifier of a semantic time series. It relies on a semantic embedding and bi-directional LSTM as a time series classifier, combined as depicted in Figure 3. In order to facilitate the understanding of our approach we will follow the step by step workflow described in this figure.

### 4.2. Step 1 and Step 2: Sensors' Stream Segmentation and Encoding

From the stream of activations recorded by the sensors, the *Step 1* consists of segmenting the dataset into event sequences. Each sequence corresponds to an activity. This segmentation is also called Explicit Window [5]. We keep the event timestamp order inside the sequences.

These event sequences are then encoded in *Step 2*. The goal of the encoding is to obtain words that can be used as inputs of the pretrained embeddings of *Step 3*. To be able to represent all possible activations of the different sensors, we create a corpus of sensors' activations and consider these activations as categorical values i.e. each sensor activation is represented by one and only one value. By transforming the sensors' activations into categorical values we allow the model to learn the notion of frequency of occurrence of an activation in a sequence. Moreover, the model, via this representation, can capture the relation of an activation to another. All these categorical variables ("words") define the smart home vocabulary that describes activities.

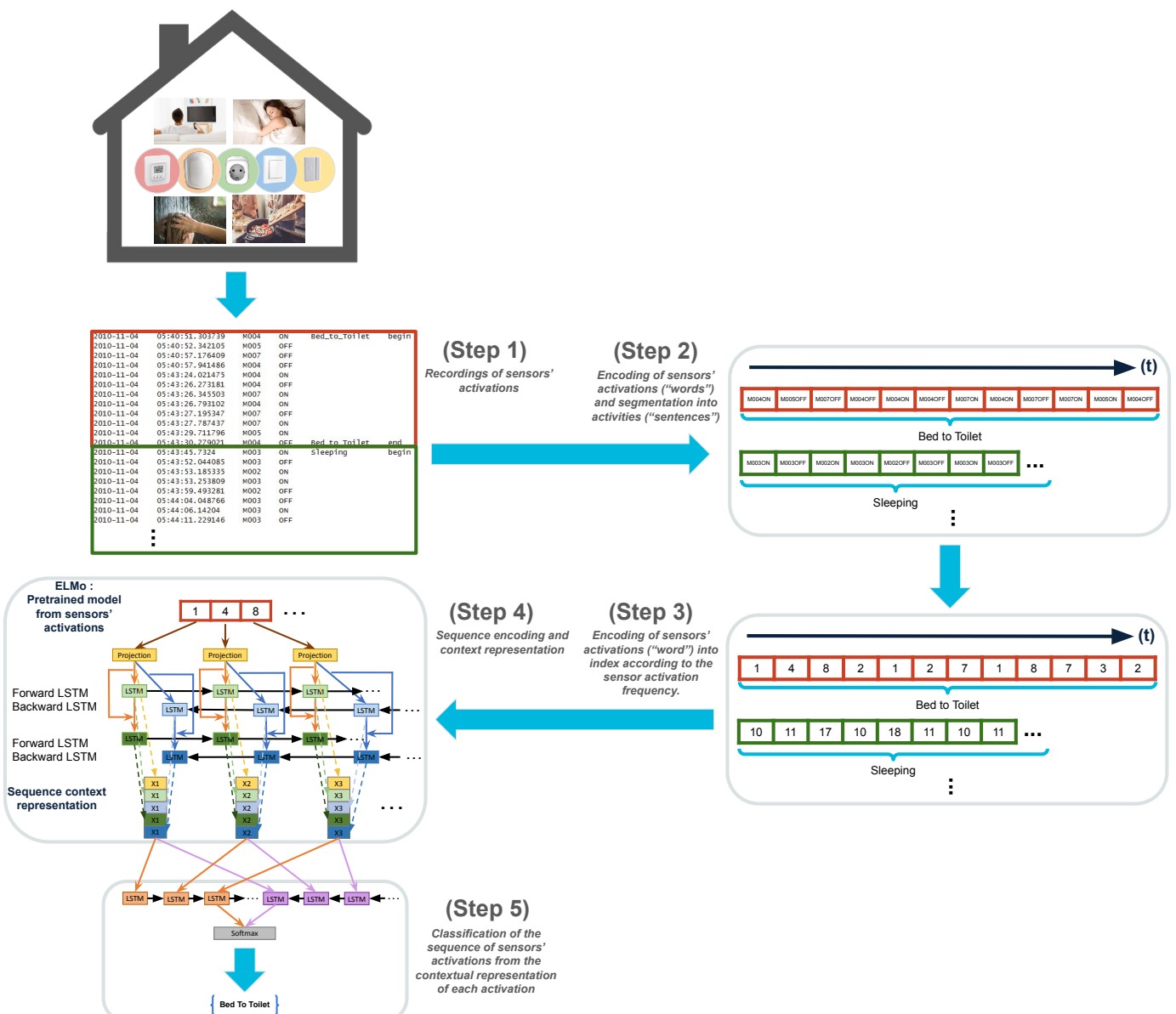

**Figure 3.** Proposed model architecture and global workflow.

More concretely, an event $e_i$ is composed of the sensor ID $s_i$, the value $v_i$ and the timestamp $t_i$. We concatenate the sensor ID $s_i$ and its value $v_i$ and ignore the timestamp $t_i$ to create the "word" associated to this event, e.g., $s_i = M001$ and the binary value $v_i = ON$ becomes M001ON. For instance the sequence of event for the activity *bed_to_toilet* in Figure 3 at Step 1 is turned into the sequence [M004ON M005OFF M007OFF M004OFF M004ON M004OFF M007ON M004ON M007OFF M007ON M005ON M004OFF]. Sensors with numerical values are represented in the same way, as a concatenation of the sensor label and the measurement. For instance, if the activation of a sensor named T004 gives a value of 24.5, the input to the system is T00424.5.

### 4.3. Step 3: Words to Indexes

As in NLP for words in sentences, each sensor activation in sequences is transformed into an index to be used as the input to a neural network. The index starts at 1 while the 0 value is reserved for the sequence padding. As in NLP, indexes are assigned following the "word" frequency (categorical value of the sensor activation frequency), e.g., if the "word" *M004ON* has the highest occurrence in the dataset, the assigned index is the lowest one i.e.,

1. This ordinal encoding encodes the frequency of a sensor activation. Thus, a sequence of "words" such as [M005OFF M007OFF M004OFF M004ON M004OFF M007ON M004ON M007OFF M007ON M005ON M004OFF] becomes the sequence of indexes [1 4 8 2 1 2 7 1 8 7 3 2], according to the sensor activation frequency, (*Step 3*).

### *4.4. Step 4: Pre-Trained Embeddings*

For each pre-segmented activity, the sequence of sensor activations is transformed into a sequence of indexes, the sequence is encoded by the pre-trained language model embedding. The workflow in Figure 3 shows the *Step 4 "Sequence encoding and content representation"* using the ELMo language model structure. Alternatively, for the Word2Vec version, the *Step 4 "Sequence encoding and content representation"* module is replaced by the Word2Vec embedding model.

The two embeddings were trained using methodologies presented in Section 3 where, sentences or sequences of words, were replaced by sequences of sensors' activations. By analogy, we consider this events stream like a text that contains sentences (the activity sequences), composed of words (the sensor activations). These embeddings were trained without supervision on the whole dataset, for each studied dataset, in order to allow models to extract features and a representation of sensor activations. ELMo predicts the activation of the next and previous sensor in the sequence while Word2Vec uses the Skip-Gram method to predict from the activation of one sensor the activations of neighboring sensors. Training is stopped when, the validation perplexity loss [28] for ELMo and the validation sparse categorical loss [29] for Word2Vec, stop decreasing. Once the embeddings were trained, their weights are frozen and finally, their output representations are used as inputs for the classifier.

### *4.5. Step 5: The ADLs Classifier*

In the *Step 5*, the classifier receives as input the encoded representation of the embedding. It can then select the features that will allow it to assign the right activity label. In our approach, we have chosen to use a bidirectional LSTM followed by a Softmax layer as classifier as proposed in [12]. Details on the parameters and how the training was conducted are provided in the next section.

## 5. Experiment

### *5.1. Datasets*

The experiment was conducted on three CASAS datasets [30]: Aruba, Milan, and Cairo, as introduced by Washington State University. Data collected from daily activities come from real apartments and houses with real inhabitants. All these living places are equipped with temperature and binary sensors, such as motion or doors sensors.

The three selected datasets are different according to the structure of the house and the number of inhabitants, see details in Table 1. Aruba is a dataset of a single person living in a house. Milan contains daily activities of one person living with a pet, while Cairo is a dataset of two persons living in the same place. They contain data of several months of labeled activities and are unbalanced, i.e., some activities are less represented than others.

We have selected these three datasets to have simple examples of activities in the case of: (1) only a single inhabitant as a baseline (Aruba), (2) a more complex dataset, when a pet can introduce more noise (Milan), and (3) a complex situation with two inhabitants (Cairo). Indeed, during the activity of one resident, a pet (in the case of Milan), or another resident (in the case of Cairo), may activate sensors that are not necessarily related to the current activity of the targeted person. These sensors activations can be considered as perturbations or noise. The algorithms must be resilient to these perturbations, in order to avoid miss-classifications. Notice that sensor activations triggered by pets are more marginal and produce less disturbances than those triggered by the other human resident. The reason is that sensors are designed to detect human activities, while pets can be out

of the field of view of the sensor or their activity can below the activation threshold of the sensors.

**Table 1.** Datasets details.

|  | Aruba | Milan | Cairo |
|---|---|---|---|
| Residents | 1 | 1 + pet | 2 + pet |
| Number of sensors | 39 | 33 | 27 |
| Number of activities | 12 | 16 | 13 |
| Number of days | 219 | 82 | 56 |

*5.2. Datasets Pre-Processing*

Liciotti et al. [12] also uses the Milan and Cairo datasets; they have groups activities under new generic labels, and more details are in Table 2. To compare our method to their work, we performed the same relabeling. The objective of this relabeling, according to the authors, is to allow a fairer comparison between the datasets in cases where same activities are labeled differently or the converse.

**Table 2.** New activity groups.

|  | Milan | Cairo |
|---|---|---|
| Bathing | Master Bathroom<br>Guest Bathroom |  |
| Bed to toilet | Bed to toilet | Bed to toilet |
| Cook | Kitchen Activity | Lunch<br>Dinner<br>Breakfast |
| Eat | Dining Room Activity |  |
| Enter home |  |  |
| Leave home | Leave home | Leave Home |
| Personal hygiene |  |  |
| Relax | Read<br>Watch Tv |  |
| Sleep | Sleep | R1 sleep<br>R2 sleep |
| Take medicine | Eve Meds<br>Morning Meds | R2 take medecine |
| Work | Desk Activity<br>Chores | Laundry<br>R1 work in office |
| Other | Meditate<br>Master Bedroom Activity<br>Other | Night wandering<br>R2 wake<br>R1 wake<br>Other |

This relabeling uses has the effect to re-balance almost the datasets, but it also, paradoxically, increases the number of examples of the "Other" class. This class corresponds to unidentified sensors activation or unidentified activation sequences. Since this class represents more than 50% of the dataset, a bias can be underlined. Indeed, if the classification algorithm is able to find all the elements of this "Other" class, then the accuracy will be at least 50%. This last aspect should be kept in mind when results are analyzed.

Contrary to Liciotti et al. [12]'s original work, we have first cleaned the datasets. Indeed, after a detailed analysis of the datasets, we noticed, especially on the Milan dataset, that the datasets contained anomalies: (1) the datasets may contain duplicate

data; (2) complete or part of days can be duplicated, e.g., in the Milan dataset; and (3) the sensors' activations may not correctly ordered temporarily, i.e., in the chronological order of timestamps.

It is also necessary to annotate each event with an activity label, paying attention to the beginning and end of the activities. Activities in the datasets are tagged with a keyword "begin" or "end" to determine when an activity starts and when it ends. However, activities can be encapsulated in other, i.e., an activity starts with the keyword "begin"; then, a few events later, a new activity starts without the previous activity being terminated by the keyword "end". Therefore, it is important to pay attention to these particular cases when pre-segmenting the dataset into activity sequences.

We observed by reproducing the work of Liciotti et al. [12] that this cleaning and annotation had an impact on the final results. We observed a loss of 5 points of accuracy on the Milan dataset using the bidirectional LSTM model of Liciotti et al. [12]. This loss is explained by a decrease in the number of occurrences of the "Other" class.

### 5.3. Training and Evaluation

In this experiment, the standard stratified K-fold cross validation method [31] was chosen. This choice is motivated by our intention to compare our results to other research studies addressing this problem (Liciotti et al. [12]). This method consists, after having segmented the dataset into sequences of activities, of distributing the sequences of activities by stratified sampling in K-folds; here, it is 3, as in Figure 4. Each fold contains 33% of the dataset. The stratified folds are obtained by preserving the percentage of samples for each class, i.e., each class of activity is present in each fold. From these K-folds, K runs are made. Each run uses K-1 folds as the training set and the last fold as the test set. During the training phase, 20% of the training set is used for the validation in order to follow and stop the training of the models before overfitting. Results, reported in our study, show the average of scores obtained on test sets.

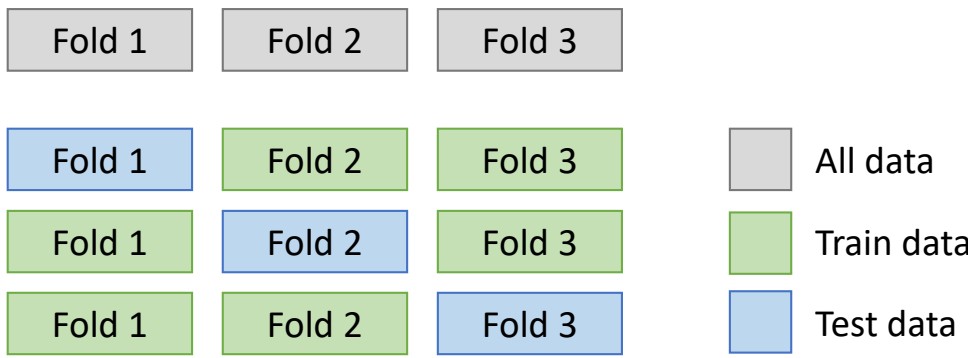

**Figure 4.** K-fold cross validation principle.

In order to speed up the training time, we have defined a maximum number of possible training epochs; in our case, 400 seem quite sufficient. In addition, the training can be stopped automatically by the system if the validation loss is not decreasing anymore; we defined 20 patience epochs. It is the well-known "early stop" method [32]. The validation loss is the metric tracked for classifiers to interrupt the training, while, for the ELMo embedding training, the metric tracked is perplexity [28]. Perplexity is used to evaluate language models in NLP. It indicates the variability of a prediction model. Lower perplexity corresponds to lower entropy and, thus, better performance. The experiment's parameters are detailed in Tables 3–5.

**Table 3.** General hyperparameters.

| | |
|---|---|
| K-fold cross validation | 3 |
| Max sequence length | 2000 |
| Max epochs number | 400 |
| Batch size | 64 |
| Patience | 20 |

**Table 4.** Embeddings' hyperparameters.

| | **Embedding** | **Word2Vec** | **ELMo** |
|---|---|---|---|
| Embedding size | 64 | 64 | 64 |
| Context windows size | None | 20 | 60 |
| Max epochs number | 400 | 100 | 400 |
| Batch size | None | None | 512 |

**Table 5.** Classifiers' hyperparameters.

| | **LSTM** | **Bi-LSTM** |
|---|---|---|
| Nb Units | 64 | 64 |
| NB layers | 1 | 1 |

We recall that the used datasets are unbalanced. Therefore, it is important to observe not only the accuracy metric and the usual F1-score but also other weighted metrics [5]. This is why we introduce the use of metrics weighted by the class support, such as the balance accuracy, the weighted precision, the weighted recall, and the weighted F1-score.

### 5.4. Hardware and Software Setup

Experiments were conducted on a server, with an Intel(R) Xeon(R) CPU E5-2640 v3 2.60 GHz, with 32 CPUs, 128 Go of RAM and a NVIDIA Tesla K80 graphic card. We did not carry out precise measurements because this is not the objective of our study. However, we have observed that the training of the algorithms on this platform takes only a few minutes: 1–2 min for the classifiers and between 10 and 40 min for the embeddings.

Keras and Tensorflow frameworks were used for the implementation of the algorithm. The Word2Vec algorithm was trained thanks to the Gensim library [33]. We used the "early stop" method provided by the Tensorflow framework. The Word2Vec training was stoped after a maximum number of epochs, see Table 4, as far as the Gensim library does not provide this method.

## 6. Results and Discussion

In this section, we will firstly try to observe what the Word2Vec and ELMo embeddings learned in an unsupervised way. Secondly, we will compare our Word2Vec and ELMo approach with the previous work done by Liciotti et al. [12], thanks to the scores obtained and the confusion matrices. Then, we will compare the ELMo approach to an extension of Liciotti et al. [12]'s model by adding an additional layer of bidirectional LSTM. Indeed, the ELMo model can be approximated by an embedding layer, followed by two tacked bidirectional LSTM layers. Finally, we will evaluate the transfer learning capability of a pretrained ELMo embedding. We used one of the three pretrained ELMo embeddings, here, Aruba, and trained a bidirectional LSTM classifier to perform the classification on the Cairo dataset.

### 6.1. Word2Vec Embedding Features

In order to visualize the embedding of the Word2Vec model, we use the UMAP algorithm [34] to reduce the dimension of the embedding vectors from dimension 64 to two dimensions. The visualization of these 2D vectors that represent each sensor activation reveals different clusters (see Figure 5a).

We have associated to each point a color representing the room where the sensor is located (Figure 5b). By observing the color taken by the clusters, it appears that, in general, each cluster corresponds to a room of the house. A zoom on the violet cluster, "kitchen", is provided in Figure 6a and can be compared to the floor map of the house in Figure 6b. All sensors on the floor map can be seen in the cluster. The two clusters located in the middle of Figure 5b are the exceptions. The two multi-color clusters are actually composed only by temperature sensors activations. The binary sensors, such as the motion or door opening sensors, belong to clusters corresponding to the house's rooms. This grouping into rooms is quite coherent because the ADLs are generally located in a sole room of the house. For instance, the activity of cooking is carried out in the kitchen, so it is normal that only sensors belonging to the kitchen are activated during this activity. It is important to note that, in an unsupervised way, the Word2Vec method has captured the notion of sensor localization by grouping the sensors belonging to the same room. This sensor localization clustering was also observed by Singla and Bose [35] in their work on IoT devices identification.

Moreover, the nature of the sensors is captured, since sensors of the same nature, such as temperature sensors, are also grouped together.

Here, we illustrate only the case of the Aruba dataset. However, generally, we have very similar results on all evaluated datasets in this experiment.

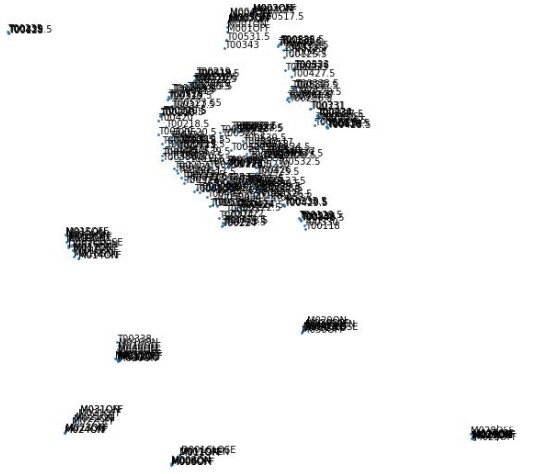

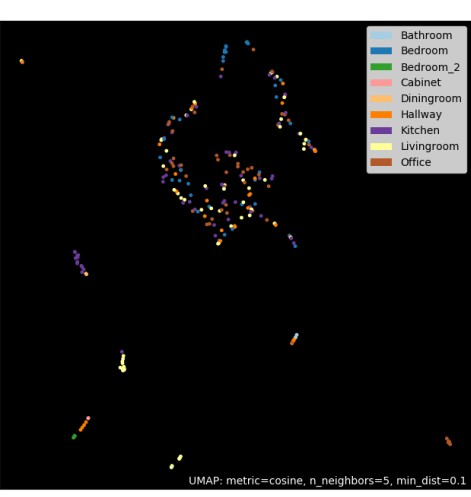

(**a**) Aruba: word embedding visualization

(**b**) Aruba: room color clusters

**Figure 5.** Aruba: Word2Vec embedding.

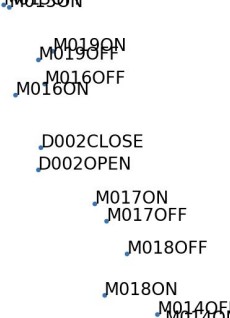

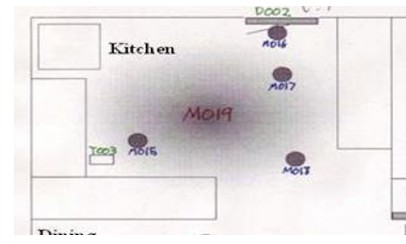

(**b**) Kitchen sensors on Aruba house map

(**a**) Word2Vec zoom on violet "Kitchen" cluster

**Figure 6.** Cluster meaning.

Aiming at visualizing the embedding of the activation sequences, we add on top of the Word2Vec model a Global Average Pooling layer [36] in order to transform our word vector sequence into a single vector. Once the set of activation sequences is transformed into a vector, we use UMAP to reduce each of the activity vectors to two dimensions. Each

activity vector is then displayed and tagged with a color corresponding to the activity label (see Figure 7).

We can observe that activities with the same label are grouped together. The activities corresponding to the "Other" class are mainly concentrated in the center of the representation, while the other activity classes are found at the periphery. Very few distinct clusters appear. All the clusters are connected by the points corresponding to the activation sequence labeled "Other".

This representation allows us to assert that Word2Vec is able to extract features able to classify the activation sequences. However, the Word2Vec embedding method does not seem to be efficient enough to isolate in individual clusters all the activity classes.

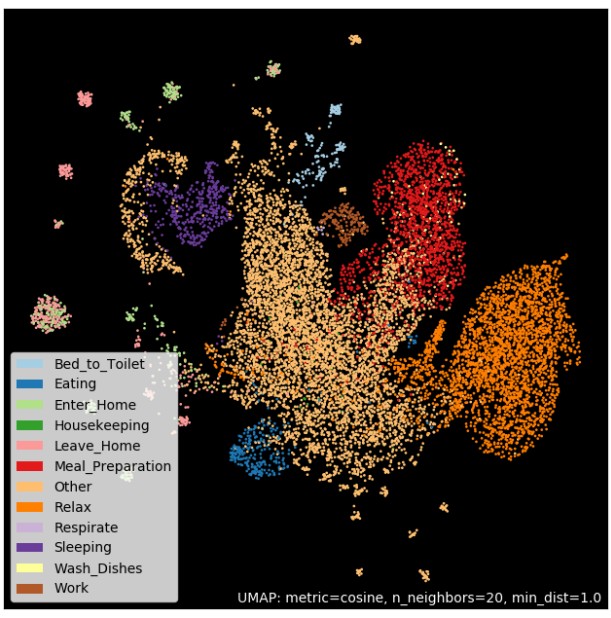

(**a**) Activity sequences of Aruba embedded by Word2Vec

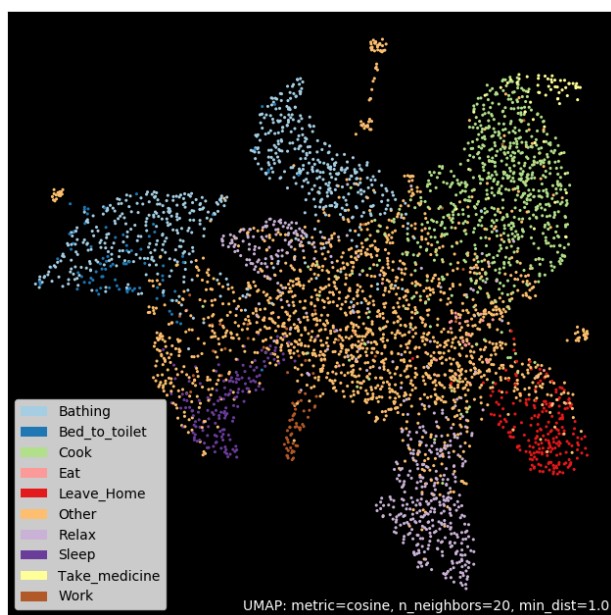

(**b**) Activity sequences of Milan embedded by Word2Vec

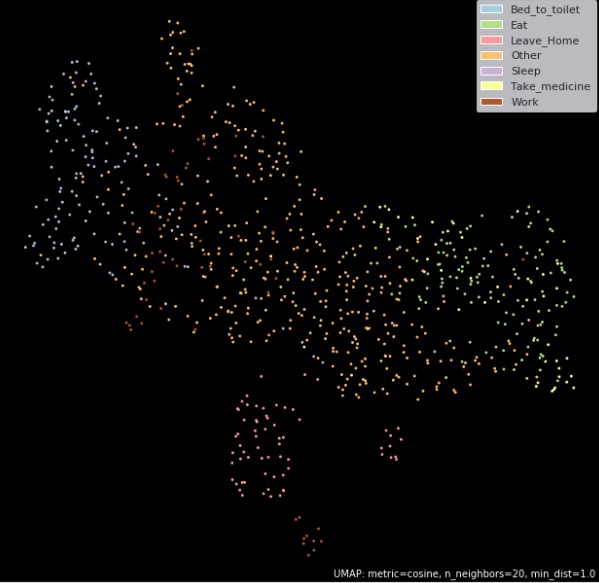

(**c**) Activity sequences of Cairo embedded by Word2Vec

**Figure 7.** Word2Vec activity sequences embedding.

### 6.2. ELMo Embedding Features

The main advantage of ELMo is to provide more than one representation for each word based on the context in which it appears. Therefore, it is a non-sense to visualize ELMo embedding of isolated words, insofar as the word vector provided by this embedding depends on surrounding words. However, the visualization of the sentence embedding garners the appearance of some interesting cues. In order to visualize the embedding of the activation sequences, we proceeded in the same way as for the Word2Vec method (see Figure 8).

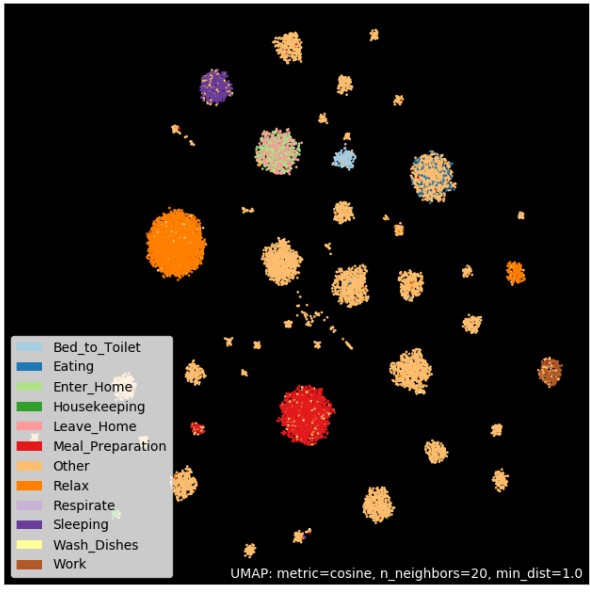

(**a**) Activity sequences of Aruba embedded by ELMo

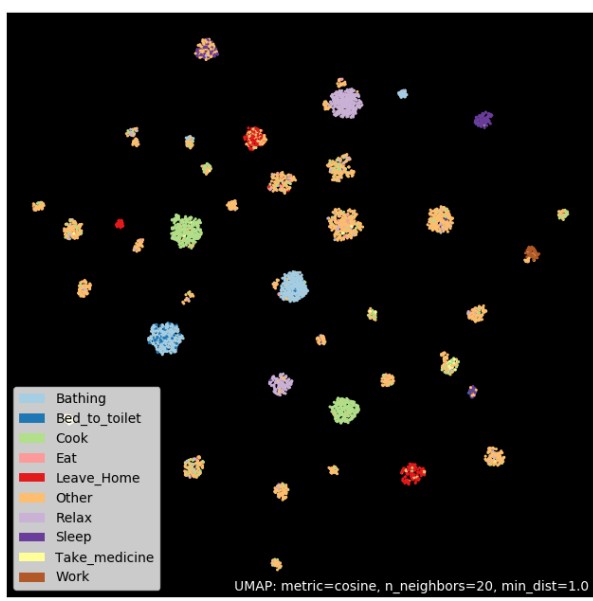

(**b**) Activity sequences of Milan embedded by ELMo

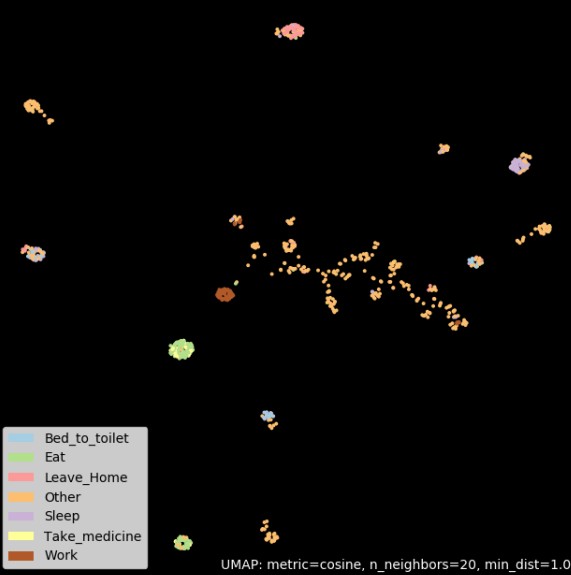

(**c**) Activity sequences of Cairo embedded by ELMo

**Figure 8.** ELMo activity sequences embedding.

Compared to the Word2Vec method, the clusters proposed by the association of ELMo and UMAP are more isolated from each other. This means that ELMo is able to extract more revealing features than Word2Vec. Moreover, within the activity class "Other", clusters appear. We assume that these clusters of activation sequences, labeled "Other", are in fact potential new activity classes. ELMo mixed with UMAP is a possible way to discover new

activities. This visualization demonstrates the strong ability of the ELMo embedding to capture relevant features from raw sensors activations.

The combination of the ELMo pretrained embedding and a dimension reduction algorithm, such as UMAP, seems able to make unsupervised clustering of similar activity sequences. Indeed, the training of the ELMo embedding and UMAP are done in an unsupervised way, i.e., by using no labels. Therefore, it seems possible to group pre-segmented activity sequences completely unsupervised. In other words, if it is possible to split a dataset into unlabeled activity sequences and, by using ELMo and UMAP, seems able to group sequences belonging to the same class.

However, this clustering is not perfect, when looking closely at the clusters. Some points of a different color than the cluster majority color can appear. This confusion can be explained in two ways. First, some sequences belonging to two different classes may be very similar in terms of sensor activation because there is not enough sensor to distinguish them. Second, it is potentially a labeling error in the dataset. Indeed, it is difficult to create and label datasets about human activities in houses [5].

### 6.3. Comparison with Context-Free Embeddings

To evaluate our proposed method, we compare our results to the work of Liciotti et al. [12]. Their works have showed that LSTM and bidirectional LSTM with an embedding layer performs well on the classification problem of sequences of sensors' activations. We also add two models for comparison, a LSTM and a bidirectional LSTM without embedding layer, to evaluate the improvement of this layer.

The experiment results in Tables 6 and 7 show that the bidirectional LSTM obtained better results, as shown in Reference [12]. In addition, using an embedding layer improved the global classification performance, thanks to the ability of this layer to capture similarity features between sensors' activations.

Against all odds, we notice that the Word2Vec embedding did not improve the classification performance compared to standard embeddings. On the contrary, using Word2Vec decreases the performance, except on the dataset Aruba, where it performs better for balance accuracy score and the weighted recall score. This means that the Word2Vec embedding did not accurately capture some important features by itself but allows a very small gain in different classes retrieval.

The ELMo embedding performs better than all other methods on every dataset, especially on the multi-user dataset Cairo. It increases the F1-score by 5 points and the weighted F1-score by 10 points. We suppose that a multi-user dataset requires the understanding of the context in order to differentiate users' activities. Moreover, as it is possible to see on the confusion matrix in Figures A1–A3, ELMo allows for finding a larger number of classes.

Concerning the training time, we noted a negligible gain of about 30 s to 2 min, depending on the dataset, by using a pretrained embedding.

**Table 6.** Without and with embedding coupled with a LSTM Classifier.

| | Aruba | | | | Milan | | | | Cairo | | | |
|---|---|---|---|---|---|---|---|---|---|---|---|---|
| | No Embedding | Liciotti | W2V | ELMo | No Embedding | Liciotti | W2V | ELMo | No Embedding | Liciotti | W2V | ELMo |
| Accuracy | 94.63 | 96.50 | 96.6 | **96.61** | 79.18 | **89.88** | 86.31 | 87.29 | 74.82 | 81.44 | 79.31 | **82.62** |
| Precision | 93.56 | 96.11 | 96.21 | **96.41** | 78.84 | **88.51** | 86.08 | 87.26 | 72.08 | 79.64 | 73.6 | **82.77** |
| Recall | 94.06 | 95.5 | 96.6 | **96.61** | 79.18 | **88.99** | 86.31 | 87.29 | 74.82 | 81.44 | 79.31 | **82.62** |
| F1-score | 93.77 | **96.88** | 96.32 | 96.43 | 78.42 | **88.63** | 85.79 | 87.07 | 72.73 | 80.09 | 75.79 | **82.05** |
| Balance Accuracy | 70.00 | 79.52 | **80.11** | 79.39 | 61.88 | 74.26 | 69.33 | **75.35** | 60.46 | 69.98 | 62.91 | **72.58** |
| Weighted Precision | 71.84 | 80.50 | 79.39 | **86.24** | 76.22 | 79.17 | 81.75 | **86.16** | 60.73 | 68.88 | 55.31 | **78.10** |
| Weighted Recall | 70.00 | 79.52 | **80.07** | 79.39 | 61.89 | 74.26 | 69.33 | **75.35** | 60.46 | 69.98 | 62.90 | **72.358** |
| Weighted F1 score | 70.55 | 79.49 | 79.34 | **80.73** | 65.94 | 76.28 | 72.07 | **78.90** | 59.11 | 68.80 | 58.92 | **73.78** |

**Table 7.** Without and with embedding coupled with a Bi-LSTM Classifier.

| | Aruba | | | | Milan | | | | Cairo | | | |
|---|---|---|---|---|---|---|---|---|---|---|---|---|
| | No Embedding | Liciotti | W2V | ELMo | No Embedding | Liciotti | W2V | ELMo | No Embedding | Liciotti | W2V | ELMo |
| Accuracy | 95.01 | 96.52 | 96.59 | **96.76** | 82.24 | **90.54** | 88.33 | 90.14 | 81.68 | 84.99 | 82.27 | **89.12** |
| Precision | 94.69 | 96.11 | 96.23 | **96.43** | 82.28 | 90.08 | 88.28 | **90.2** | 80.22 | 83.17 | 82.04 | **88.41** |
| Recall | 95.01 | 96.50 | 96.59 | **96.69** | 82.24 | **90.45** | 88.33 | 90.31 | 81.68 | 82.98 | 82.27 | **87.59** |
| F1-score | 94.74 | 96.22 | 96.32 | **96.42** | 81.97 | 90.02 | 87.98 | **90.1** | 80.49 | 82.18 | 81.14 | **87.48** |
| Balance Accuracy | 77.73 | 79.96 | **81.06** | 79.98 | 67.77 | 74.31 | 73.61 | **78.25** | 70.09 | 77.52 | 69.38 | **87.00** |
| Weighted Precision | 79.75 | 82.30 | 82.97 | **88.64** | 79.6 | 82.03 | 84.42 | **87.56** | 68.45 | 80.03 | 77.56 | **86.83** |
| Weighted Recall | 77.73 | **80.71** | 81.06 | 79.17 | 67.77 | 75.51 | 73.62 | **78.75** | 70.09 | 73.82 | 69.38 | **84.78** |
| Weighted F1 score | 77.92 | 81.21 | 81.43 | **81.93** | 71.81 | 77.74 | 76.59 | **82.26** | 68.47 | 74.84 | 70.95 | **84.71** |

## 6.4. Confusion Matrices Analysis

The confusion matrices in Appendix A allow us to visualize the classification rate for each class of the embedding + bidirectional LSTM and ELMo + bidirectional LSTM approaches. We observe that the misclassification rate is lower with ELMo method. This rate is almost two times smaller on the Cairo dataset, a multi-user dataset.

However, for the Aruba dataset, we observe that the activity "Respirate" is not correctly recognized for the both methods. This activity has very few examples (5 examples) and, therefore, is difficult to recognize. The activity "Wash Dishes" is poorly recognized and is very confused with the activity "Meal Preparation". This confusion is explained by the fact that these two activities activate the same sensors. In order to solve this problem, the algorithms should be able to take into account the time of day, or the activity preceding the current activity.

In the case of the Milan dataset, the activity "Eat" is also poorly recognized because it mainly activates only the motion sensor located in the dining room. However, our method using the ELMo embedding is able to find a part of the occurrences of this activity. This performance is explained by the fact that ELMo captures in which context the motion sensor in the dining room is activated. In other words, it is able to differentiate between the resident's passage in the dining room to join the kitchen or the living room, from a real activation of the sensor for the "Eat" activity.

## 6.5. Comparison Against Staked Bidirectional LSTM

The ELMo model can be approximated by a bidirectional LSTM layer with an embedding layer. In this experiment, we compare the ELMo method to two stacked layers of bidirectional LSTM with an embedding layer. Table 8 shows the results of this comparison.

The results demonstrate that the ELMo structure obtains a non-negligible gain on the three datasets, except on the Milan dataset. On this one, gains are less, but not negligible, on the weighted scores. ELMo still allows recognizing more classes and is more precise than the other structures. The stacking of bidirectional LSTM does not allow obtaining better performance. This type of structure can even degrade the performance in some cases, as on the Cairo dataset, or Aruba. It seems that the method to train ELMo helps to capture more useful features.

**Table 8.** Comparison with two layers of Bi-LSTM.

| | Aruba | | | Milan | | | Cairo | | |
|---|---|---|---|---|---|---|---|---|---|
| | 1 L | 2 L | ELMo | 1 L | 2 L | ELMo | 1 L | 2 L | ELMo |
| **Accuracy** | 96.52 | 96.46 | **96.76** | **90.54** | 90.03 | 90.14 | 84.99 | 84.99 | **89.12** |
| Precision | 96.11 | 96.04 | **96.43** | 90.08 | **90.22** | 90.20 | 83.17 | 85.04 | **88.41** |
| Recall | 96.50 | 96.41 | **96.69** | **90.45** | 90.28 | 90.31 | 82.98 | 84.4 | **87.59** |
| F1-score | 96.22 | 96.13 | **96.42** | 90.02 | 90.07 | **90.10** | 82.18 | 84.08 | **87.48** |
| Balance Accuracy | 79.96 | 78.74 | **79.98** | 74.31 | 75.51 | **78.25** | 77.52 | 76.52 | **87.00** |
| Weighted Precision | 82.30 | 82.01 | **88.64** | 82.03 | 84.29 | **87.56** | 80.03 | 80.87 | **86.83** |
| Weighted Recall | **80.71** | 79.05 | 79.17 | 75.51 | 77.31 | **78.75** | 73.82 | 76.6 | **84.78** |
| Weight F1-score | 81.21 | 79.97 | **81.93** | 77.74 | 79.29 | **82.26** | 74.84 | 77.44 | **84.71** |

### 6.6. Transfer Learning

In the field of NLP, embeddings are pretrained on large corpora and then used on a more specific corpus to perform particular tasks, such as text classification. This is the principle of transfer learning. The objective is to use the very generic features of the large corpus on a smaller and specific corpus to gain in learning time but also in genericity.

In the case of ADLs recognition, this practice would allow for transfer of the knowledge from a smart house to another one so that the latter can recognize ADLs without further training. This transferred model could then be refined for the context of this new house. We experimented with this practice using Aruba's ELMo embedding on a the Cairo dataset.

To do this, we trained the ELMo model on the dataset Aruba. Then, we used this trained model to extract and encode the activity sequence frames of the Cairo dataset. These features are then given as input to a classifier. The classifier is a neural network composed of a bidirectional LSTM, followed by a softmax layer. The weights of the ELMo embedding were frozen, and only the bidirectional LSTM and the softmax were trained to classify the activities of the second dataset. The results of the experimentation can be seen in Table 9.

These results show that the generic features learned on the Aruba dataset allowed the classification of activities in the Cairo dataset with scores equivalent to the ELMo embedding trained on Cairo. We assume that the Aruba ELMo embedding was able to capture enough features about the "syntax" and the order of activation of the sensors, as well as the nature of the activated sensor, to encode the activation sequences efficiently, despite a different vocabulary.

Indeed, the two datasets do not have the same number of sensors. The name of the sensors is also different in some cases. The set of word is different from one house to another, but, in our case, both Aruba and Cairo datasets belong to the CASAS datasets, which use the same sensor type and the same denomination structure. The sensors follow the structure: "sensor type" + index. However, this experiment could not have worked fully if the vocabulary was too different because of out of vocabulary cases. We observe that, even if a word does not have the same "meaning" from one dataset to another (for example, "M001ON" corresponds to the motion sensor in the kitchen in dataset "A" and to the motion sensor in the bathroom in dataset "B"), the encoding provided by the ELMo embedding generate patterns still allows the classifier to reach performance rates equivalent to a model fully trained on the destination dataset. We conjecture this good performance comes from the fact that ELMo takes into account the word order. Thus, even if the input words have changed, the syntax is captured by the classifier, i.e., the order of each word, as well as the patterns of their recurrence in the sequence. These results indicate the importance of contextualized embeddings.

**Table 9.** Comparison between ELMo trained on the Cairo dataset and ELMo trained on the Aruba dataset, applied on the Cairo dataset (Bidirectional LSTM classifier).

|  | Cairo | |
|---|---|---|
|  | **ELMo from Cairo** | **ELMo from Aruba** |
| Accuracy | 89.12 | **89.24** |
| Precision | **88.41** | 87.77 |
| Recall | **87.59** | 86.35 |
| F1-score | **87.48** | 85.88 |
| Balance Accuracy | **87.00** | 84.02 |
| Weighted Precision | 86.83 | 87.55 |
| Weighted Recall | **84.78** | 79.56 |
| Weighted F1-score | **84.71** | 80.80 |

## 7. Conclusions and Discussion

Human activity recognition is a very dynamic and challenging research area that plays a crucial role in various applications, especially for smart homes. Such IoT environments

require robust activity learning technology to provide adequate services to the residents. The topology of homes, their different sensor and actuator installations, and the different lifestyle habits of residents add variability of the sensors' data, while IoT data are sparse. Thus, the activity modeling is challenging. It is not only a problem of pattern recognition but also a spatio-temporal sequence analysis problem, where the semantics and context of each sensor trigger can change the meaning of a sensor activation. Moreover, the nature of the activated sensor can give a certain amount of information about the current activity.

In this study, we proposed a new approach, applied for the first time to the field of recognition of activities of daily living in smart homes. We used techniques from the domain of NLP to capture the context and the semantics of sensor activations in an embedding. This approach allows recognition of a larger number of activity classes, despite the fact that datasets remain unbalanced. Indeed, fewer activation sequences are confused with the "Other" class, which, nevertheless, represents more than 50% of the datasets.

The visualization of the Word2Vec embedding allowed us to realize that this method has captured some relations between the activations of sensors. It appears that sensors of the same nature have a close distance in this embedding space. Moreover, the clusters that appear in the space represent different rooms in the house.

Our experimentation shows that capturing the context of a sensor activation allows for improvement of the classification of activity sequences, particularly on datasets containing activities performed by several residents or that became noisy by pets.

Finally, we were able to evaluate that an embedding trained in a house could be reused in a new environment containing another denomination of sensors and allow a high rate of classification of activities. It should be noted that these methods are capable to extract generic information, transferable to other datasets. This last observation suggests that transfer learning between environments is possible through these methods, as it is possible today in the NLP domain.

Through our proposition, that combines a language model embedding the semantics of sensor activations and a time-series classification algorithm, our experimental results on real smart home data highlight the importance of a dynamic contextualized semantic representation in ADL recognition. Moreover, our results show that such a representation can be shared across datasets to allow transfer learning. These findings could be the key to solve the main problems of smart home data: the scarcity and the variability that hinder any possible generalization of ADL recognition models.

In a future work, we plan to apply unsupervised learning methods based on transformers, such as BERT [22] or GPT [23]. Indeed, transformers have become state-of-the-art in the NLP domain, thanks to their ability to capture distant dependencies and to focus attention on important elements in sequences. Our goal, even via these transformer-based structures, remains the same: to capture a broader context but also to use more advanced tokenization methods to take into account the activation of unknown sensors. It should be noted that the method proposed here is limited by a certain size of vocabulary or possible sensor activation. It is currently impossible to obtain a representation of sensor values that have never been observed. Methods, such as byte pair encoding (BPE) [37] or WordPiece [38], could be considered splitting words or tokens, representing a sensor activation, into subwords or word compositions. This should capture more semantics addressing the construction of these words, thus taking into account new activation values.

**Author Contributions:** Writing—original draft, D.B.; Writing—review and editing, S.M.N., C.L., B.L. and I.K. All authors have read and agreed to the published version of the manuscript.

**Funding:** The work is partially supported by project VITAAL and is financed by Brest Metropole, the region of Brittany and the European Regional Development Fund (ERDF). This work was carried out within the context of a CIFRE agreement with the company Delta Dore in Bonemain 35270 France, managed by the National Association of Technical Research (ANRT) in France.

**Conflicts of Interest:** The authors declare no conflict of interest.

## Appendix A. Confusion Matrices

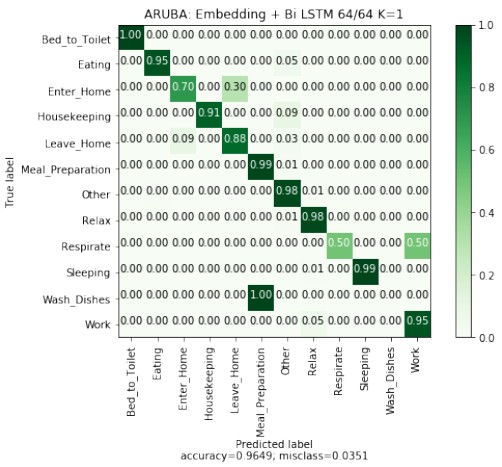

(**a**) Aruba Liciotti (Embedding + Bi-LSTM) K = 1

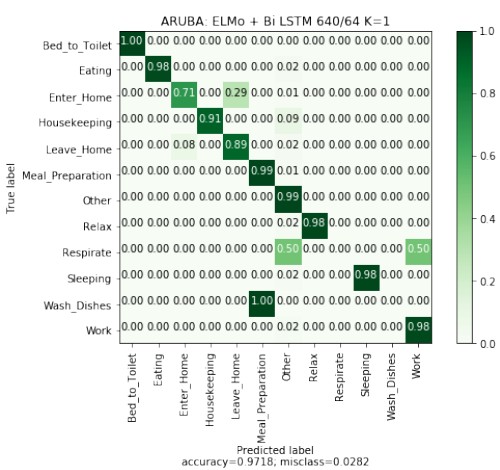

(**b**) Aruba ELMo + Bi-LSTM K = 1

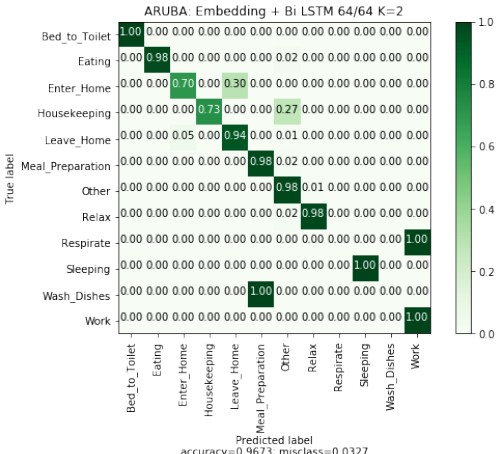

(**c**) Aruba Liciotti (Embedding + Bi-LSTM) K = 2

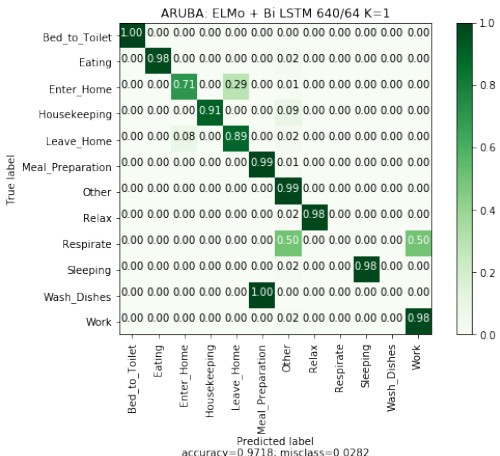

(**d**) Aruba ELMo + Bi-LSTM K = 2

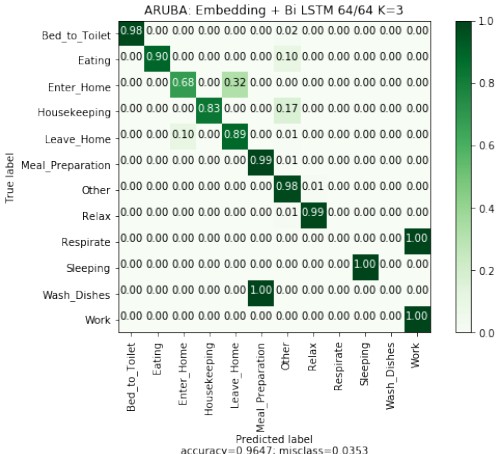

(**e**) Aruba Liciotti (Embedding + Bi-LSTM) K = 3

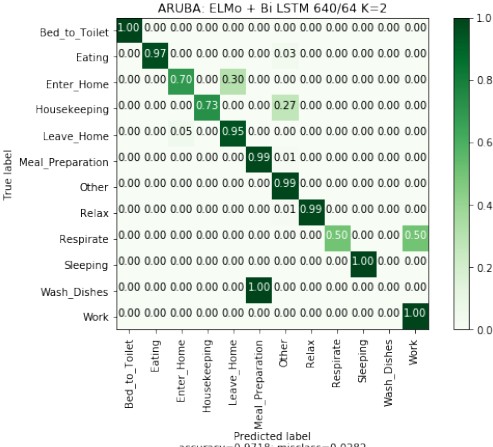

(**f**) Aruba ELMo + Bi-LSTM K = 3

**Figure A1.** Aruba confusion matrices.

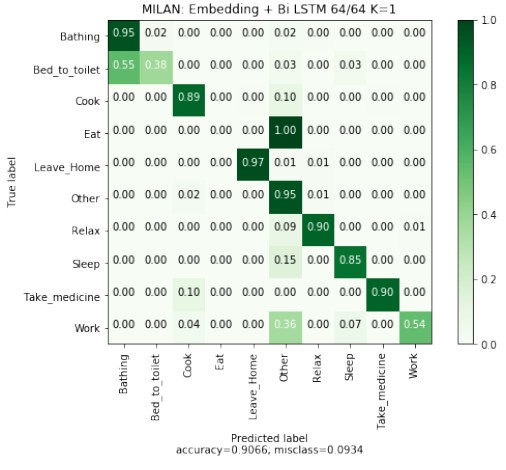

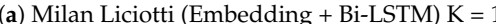

**(a)** Milan Liciotti (Embedding + Bi-LSTM) K = 1

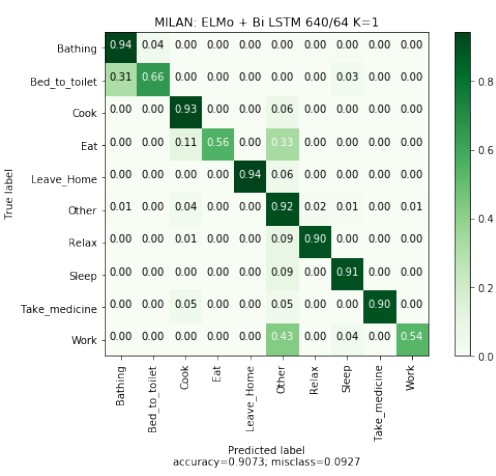

**(b)** Milan ELMo + Bi-LSTM K = 1

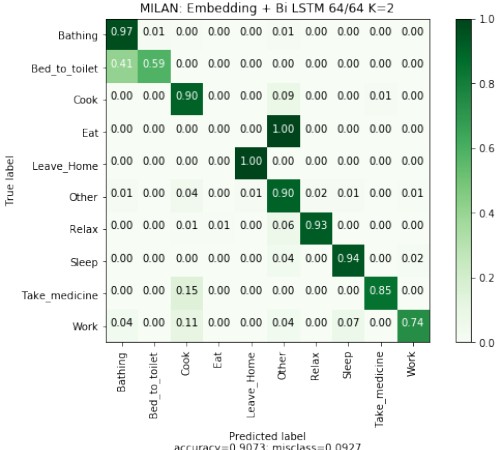

**(c)** Milan Liciotti (Embedding + Bi-LSTM) K = 2

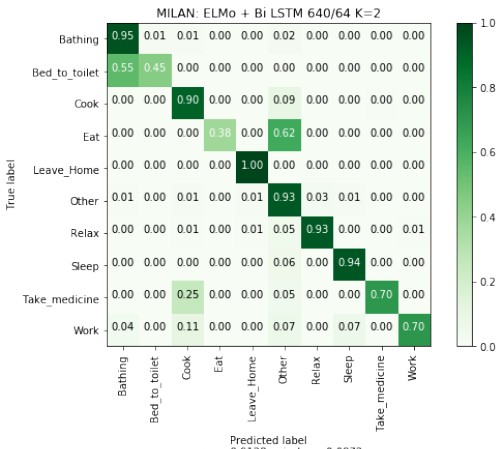

**(d)** Milan ELMo + Bi-LSTM K = 2

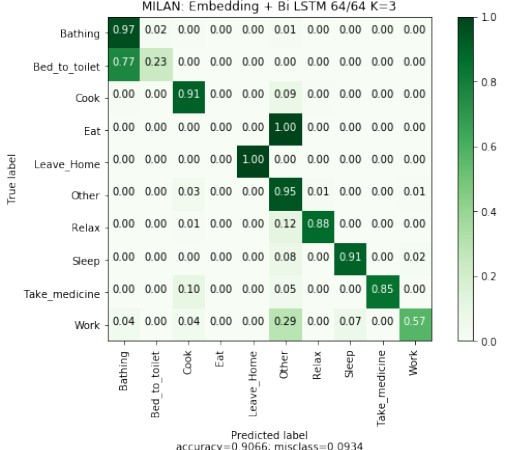

**(e)** Milan Liciotti (Embedding + Bi-LSTM) K = 3

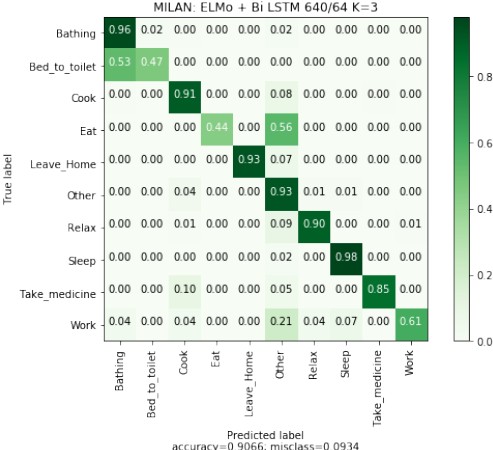

**(f)** Milan ELMo + Bi-LSTM K = 3

**Figure A2.** Milan confusion matrices.

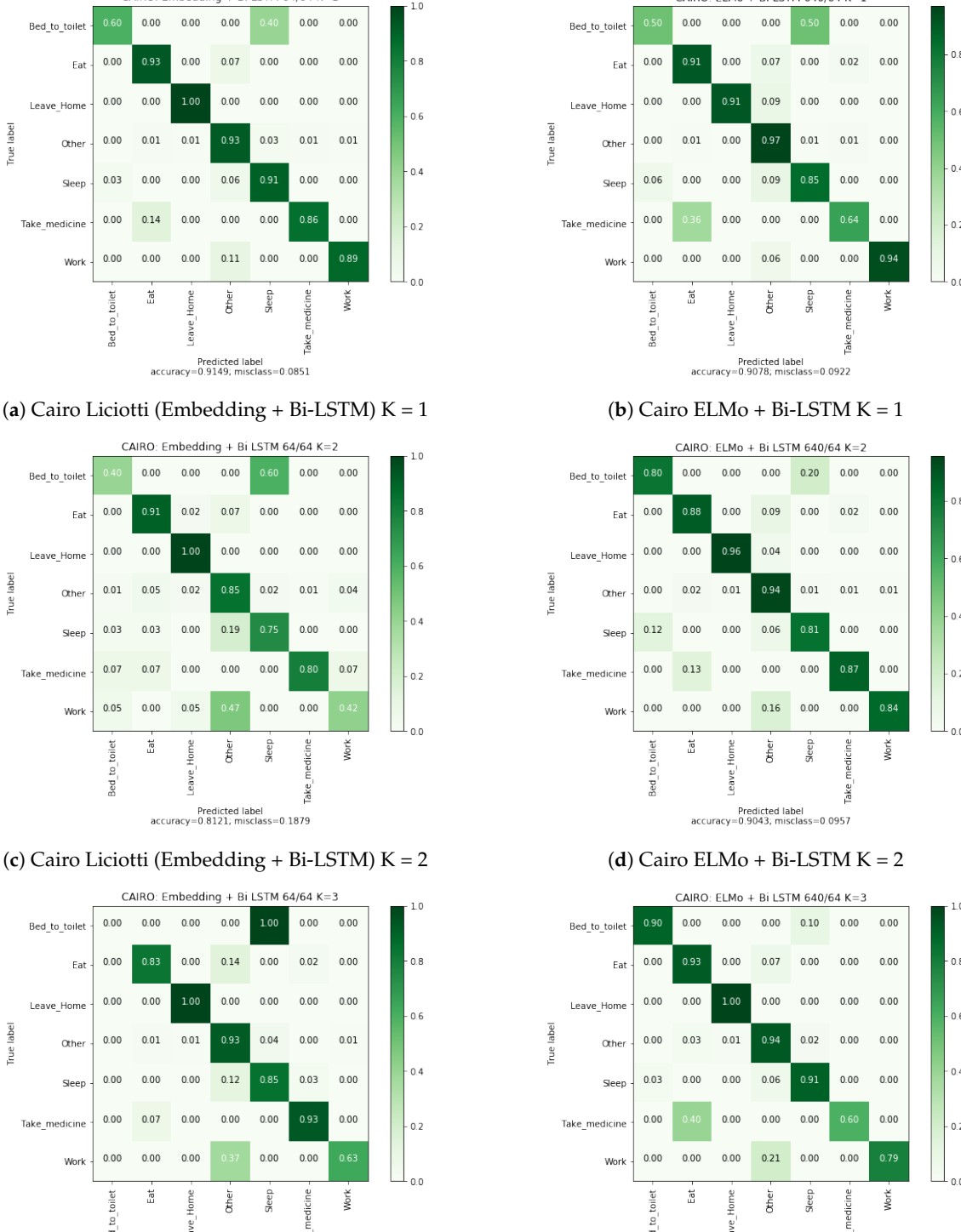

(**a**) Cairo Liciotti (Embedding + Bi-LSTM) K = 1

(**b**) Cairo ELMo + Bi-LSTM K = 1

(**c**) Cairo Liciotti (Embedding + Bi-LSTM) K = 2

(**d**) Cairo ELMo + Bi-LSTM K = 2

(**e**) Cairo Liciotti (Embedding + Bi-LSTM) K = 3

(**f**) Cairo ELMo + Bi-LSTM K = 3

**Figure A3.** Cairo confusion matrices.

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
