# Peer review of "Using Language Model to Bootstrap Human Activity Recognition Ambient Sensors Based in Smart Homes"

_electronics, doi:10.3390/electronics10202498_

Round 1
Reviewer 1 Report
Authors are suggested to address the following comments.
Comment 1. Abstract, please address:
(a) Define all abbreviations.
(b) Refer to the template, the maximum word count is 200 words.
(c) Briefly summarize the key findings/results of the proposed work.
Comment 2. More terms should be included in the keywords, for instance, long short-term memory, transfer learning, etc.
Comment 3. Section 1 Introduction, please address:
(a) 1.1, correct “such as motion or doors sensors...”
(b) Enhance the discussion on the rationale before defining the goal of the study (Line 40).
(c) Regarding the organization of Subsections 1.1 and 1.2, are they background information?
(d) Move the research contributions of the paper from 2.4 to Section 1. Authors could make clear research contributions in point-form.
Comment 4. Section 2 Related works, please address:
(a) To avoid confusion, headings of subsections 2.2 and 2.3 should include “approach”.
(b) More latest journal articles (2019-Present) should be included.
(c) The suggested information to be included is not only the methodology but also the results and limitations of existing works.
(d) Briefly compare the rationale/advantages/disadvantages of the three approaches (2.1, 2.2, and 2.3).
Comment 5. Section 3 Background and proposed approach, please address:
(a) Heading of 3 overlaps with heading of 3.1. Also, it seems that section 3 does not contain the information related to background.
(b) Formulations and pseudo-code are expected.
Comment 6. Section 4 Experimental setting, please address:
(a) The information shared in Appendix should be moved to main sections. They are important.
(b) Enhance the resolutions of all figures.
(c) Figures 2 and 3, more elaboration is needed to explain the figures.
(d) References are missing for the existing works being compared.
(e) Perform statistical analysis to confirm the proposed work outperforms existing works.
(f) Subsection 4.2, for k-fold cross validation, it is suggested to cite the following work (title as follows):
Predicting at-risk university students in a virtual learning environment via a machine learning algorithm
(g) Table 6, correct “K” to “K-fold”.
(h) Table 8, correct “Bi LSTM” to “Bi-LSTM”.
Reviewer 2 Report
The aim of the paper is to develop new methods to recognize the daily activities in smart homes. The proposed approach combines LSTMs with NLP-based techniques in order to obtain accurate algorithms.
The authors used several well-known measures to evaluate the performances of the proposed algorithms. The experimental results seem to point out that the LSTM combined with W2V and ELMO method respectively outperforms both LSTM and Bidirectional LSTM with an embedding layer techniques. Also, several methods designed to speed up the computation are provided.
This paper deals with an interesting subject, it is well written and easy to read. The methodology is clearly described. The conclusions are compelling and supported by the experiments.
In my opinion, the paper could be published, but after some minor corrections/ updates.
First, I suggest that the complexity of the algorithms, using runtime measurements for example, should be taken into account to evaluate the performances of the reported approaches.
Secondly, in several places, the text needs to be polished – lines 35, 52, 66. Also, figure 2 and 3 are quite blurred and need improvements. Figures A1 and A2 need further explanations, while Table 9 should be properly aligned.
